# SSR-Based Molecular Identification and Population Structure Analysis for Forage Pea (*Pisum sativum* var. *arvense* L.) Landraces

**DOI:** 10.3390/genes13061086

**Published:** 2022-06-18

**Authors:** Kamil Haliloglu, Aras Turkoglu, Mustafa Tan, Peter Poczai

**Affiliations:** 1Department of Field Crops, Faculty of Agriculture, Ataturk University, 25240 Erzurum, Turkey; kamilh@atauni.edu.tr; 2Department of Biology, Faculty of Science, Cankiri Karatekin University, 18200 Çankırı, Turkey; 3Department of Field Crops, Faculty of Agriculture, Necmettin Erbakan University, 42310 Konya, Turkey; 4Havsa Vocational College Park and Garden Plants, Trakya University, 22030 Edirne, Turkey; mustafatan@trakya.edu.tr; 5Botany Unit, Finnish Museum of Natural History, University of Helsinki, P.O. Box 7, FI-00014 Helsinki, Finland; 6Institute of Advanced Studies Kőszeg (iASK), 9731 Kőszeg, Hungary

**Keywords:** genetic, molecular markers, structure, UPGMA

## Abstract

Plant genetic diversity has a significant role in providing traits that can help meet future challenges, such as the need to adapt crops to changing climatic conditions or outbreaks of disease. Our aim in this study was to evaluate the diversity of 61 forage pea specimens (*P. sativum* ssp. *arvense* L.) collected from the northeastern Anatolia region of Turkey using 28 simple sequence repeat (SSR) markers. These primers generated a total of 82 polymorphic bands. The number of observed alleles (Na) per primer varied from 2 to 4 with a mean of 2.89 alleles/locus. The mean value of expected heterozygosity (Exp-Het = 0.50) was higher than the mean value of observed heterozygosity (Obs-Het = 0.22). The mean of polymorphic information content (PIC) was 0.41 with a range of 0.03–0.70. The mean number of effective alleles (Ne) was found to be 2.15, Nei’s expected heterozygosity (H) 0.49, and Shannon’s information index (I) 0.81. Cluster analysis through the unweighted pair-group mean average (UPGMA) method revealed that 61 forage pea landraces were divided into three main clusters. Genetic dissimilarity between the genotypes, calculated with the use of NTSYS-pc software, varied between 0.10 (G30 and G34) and 0.66 (G1 and G32). Principal coordinate analysis (PCoA) revealed that three principal coordinates explained 51.54% of the total variation. Moreover, population structure analysis showed that all genotypes formed three sub-populations. Expected heterozygosity values varied between 0.2669 (the first sub-population) and 0.3223 (third sub-population), with an average value of 0.2924. Average population differentiation measurement (Fst) was identified as 0.2351 for the first sub-population, 0.3838 for the second sub-population, and 0.2506 for the third sub-population. In general, current results suggest that SSR markers could be constantly used to illuminate the genetic diversity of forage pea landraces and can potentially be incorporated into future studies that examine the diversity within a larger collection of forage pea genotypes from diverse regions.

## 1. Introduction

Forage pea (*P. sativum* ssp. *arvense* L.) is a significant legume crop for fresh and dry grass production as well as an alternative to barley and vetch in animal nutrition [1]. Field-pea (*P. sativum* ssp. *arvense* L.) containing 2n = 2x = 14 chromosomes is a self-pollinated winter legume belonging to the *Fabaceae* (*Leguminosae*) family and sub-family *Papilionaceae* [2]. Nutritionally, it is a rich source of protein (21–25%) with high concentrations of lysine and tryptophan amino acids [3] and low levels of cysteine and methionine amino acids [4]. 

Forage pea plays an important role in meeting global forage demand. The annual green herbage production of forage pea is about 21 million tones on 2.7 million ha area globally, and 452.776 tons on about 24 thousand ha area in Turkey [1]. Although Turkey is the genetic pool and center of origin of many cultivated and wild forage crops, the production of high-quality hay remains a key challenge in Turkey compared to the global production scale due to insufficient forage cultivation [5], while the average productivity of forage peas in Turkey is very low. A quantum increase in total pulse output is required to increase availability per animal and to meet the feed requirements of the growing animal population. Therefore, high-yielding forage pea varieties and quality seed production are essential to the continuity of production. In any crop breeding program, there is a need to focus on the selection of key characteristics and the creation of genetic variability to improve yield and the component characteristics. However, the genetic variation available among the forage pea germplasm, the nature of the component traits on which selection will be effective, and the impact of environmental factors on each trait must be understood before effective selection can be achieved [6]. In Turkey, as all over the world, studies on natural landraces have been employed so that forage crops can contribute to agriculture, and such problems can be overcome. Landraces are valuable sources of genetic traits that are of interest to plant breeders for inclusion in breeding programs [7]. Furthermore, landrace populations make up a significant portion of crop genetic variation and are often characterized by high stress tolerance and local adaptation. [8]. The genetic variability of landrace inclusion can provide crucial information to assist gene banks replicate and conserve these genetic resources correctly [9]. In addition, information regarding genotype genetic variability aids decision-making for conservation tasks such as collecting and managing genetic resources, finding genes to add value to genetic resources, and predicting the ability to mix breeding material or quickly validate it for breeding purposes. As a result, determining the genetic variability of germplasm is the first stage in improving and developing superior cultivars, which is referred to as pre-breeding. In landrace genotypes, genetic diversity is crucial for sustainability because these varieties offer high yield stability despite low yield capacity. The genetic heterogeneity of landraces contributes to the resilience of the production system in response to biotic and abiotic stresses, reducing the overall risk of crop failure [10]. Up to now little has been known about the landrace of forage peas. Therefore, knowledge of genetic variability parameters for quantitative traits will help us leverage landraces to develop a superior variety. Several studies have been published on the pea germplasm using morphological descriptors and agronomic features, and recently DNA markers. Since many morphological characters are affected by environmental factors, the analysis of genetic diversity among local pea populations in this study was performed on molecular markers [11]. Molecular markers are useful for complementing morphological traits as they are independent of environmental influences and allow variety identification in the early stages of the crop [12]. The study of genetic diversity has received much attention in the last two decades for an effective breeding program and germplasm management, especially after the development of the concept of seed collection [13]. Several different marker systems, such as restriction fragment length polymorphisms (RFLPs), amplified fragment length polymorphism (AFLPs), simple-sequence repeats (SSRs), and inter-primer binding site (iPBS) have been used for either mapping studies or diversity assessment in pea (*Pisum sativum* L.). However, few studies have used DNA markers in forage pea (*Pisum sativum* ssp. *arvense* L.). Among molecular markers based on PCR (Polymerase Chain Reaction) methods, SSR markers are an ideal approach for characterizing large numbers of landraces in a relatively short time and at low cost. SSR markers are suitable for whole genome characterization because of their uniform distribution throughout the genome, co-dominant inheritance, multiple allelic structure, ease of detection by PCR, and good reproducibility [14]. 

Recently, Demirkol and Yilmaz [15] studied the genetic diversity of 48 forage pea landraces collected from the East Black Sea Region in Turkey. Forage pea landraces were collected using the criteria of the International Union for the Protection of New Varieties of Plants (UPOV) and used 32 SSR markers. According to these studies, the Turkish forage pea landrace collection has a significant genetic variation. Despite the fact that the research area where the landraces were collected in the current study did not cover a large area, the results revealed that the diversity results were either similar or higher than those found in studies that collected landraces from larger areas. In other words, the landraces that have been evaluated so far represent only a small portion of the available resources; also, they are limited to a few geographic zones, making it impossible to investigate the genetic structure of forage pea landraces in Turkey. Furthermore, no comprehensive study has yet been undertaken to quantify the genetic diversity of forage pea germplasm in Turkey. Therefore, the aim of this study was to characterize and evaluate the genetic diversity of forage pea genotypes collected from various provinces in the northeastern Anatolia region of Turkey using SSR markers, and to develop strategies to protect the endangered genetic resources of this region.

## 2. Materials and Methods

### 2.1. Plant Material

Sixty-one Turkish forage pea (*P. sativum* ssp. *arvense* L.) landraces were used as plant materials in the present study. The names of the landraces and their collection sites are shown in Table 1 and briefly displayed in Figure 1. The Turkish forage pea landraces were collected in cultivated fields from 5 different provinces (Erzurum, Bayburt, Ardahan, Kars, and Giresun) in the northeastern Anatolia region of Turkey.

### 2.2. Genomic DNA Extraction and Quality Assessment

Plants were grown in a greenhouse as part of the study. In the Laboratory of Molecular Biology and Genetics, Department of Field Crops, Agriculture Faculty, Ataturk University, bulk DNA of 61 individuals per accession was extracted from young leaves of 2-week-old plants. The procedure as described by Zeynalzadeh-Tabrizi et al. [16] was followed for extracting genomic DNA from three plants of each accession, which were then bulked for further analysis. Extracted DNA was electrophoresed in 0.8% agarose gel.

### 2.3. PCR Reaction and Genotyping

Initially, 28 primers developed by Xu-Xiao et al. [17], Smýkal et al. [18], Zong et al. [19], Zong et al. [20], and Cieslarová, et al. [21] were used to screen six forage pea genotypes in order to see which primers produced sharp and clear banding profiles. All tested primers with better PCR products were chosen to genotype the entire set of forage pea accessions. The names of the primers, their annealing temperatures and sequences are listed in Table 2. PCR Amplification was performed in a thermos cycler (SensoQuest Labcycler, Göttingen, Germany) and was conducted in 10 µL reaction mixture comprising 25 ng template DNA, 0.5 U Taq polymerase, 0.25 mM dNTP, 1 µM (20 pmol) primer, 10X buffer; 2 mM MgCl_2_. The PCR thermal cycling profile was as follows: initial denaturation for 3 min at 95 °C, 38 cycles of 95 °C for 60 s, 53–66 °C (annealing temperature depending on primers; for details see Table 2) for 60 s, 72 °C for 120 s, and final extension at 72 °C for 10 min [22]. At 200 V for 105 min, all PCR amplification products were separated by polyacrylamide Mega-Gel dual vertical electrophoresis (Model C-DASG-400-50). In 0.5X TBE buffer, 6% (*w*/*v*) acrylamide/bis-acrylamide (19:1), 0.07% (*w*/*v*) ammonium persulfate, and 0.08% (*w*/*v*) TEMED were used to prepare the gel solution. Finally, the gels were photographed using a digital camera (Model Nikon Coolpix500, Nikon, Japan) under UV light (Appendix A).

### 2.4. Data Analysis

The SSR band patterns were analyzed with TotalLab TL120 software. Scoring was performed for SSR amplification products as present (1) or absent (0). Power Marker version 3.25 [23] was applied to obtain information on major allele frequency (MAF), gene diversity (GD), and polymorphic information content (PIC). Observed heterozygosity (Obs-Hom), Expected heterozygosity (Exp-Het), number of effective alleles (ne), Nei’s expected heterozygosity (h), and Shannon’s information index (I) values were determined with POPGEN1.32 software [24]. The Dice similarity index [25] was used to calculate genetic similarity between each pair of accessions. The similarity matrix was then used in NTSYS-pc V2.1 to create a dendrogram using the unweighted pair group method with the arithmetic mean (UPGMA) and SAHN clustering [26]. GenAlExV6.5 software was used for principal coordinate analysis (PCoA) and analysis of molecular variance (AMOVA) [27]. Genetic structure of the genotypes was assessed through model-based cluster analysis with Structure v. 2.2 software [28]. The number of population clusters (K) was estimated using Evanno’s ΔK method [29] and Structure Harvester [30]. MCMC (Markov Chain Monte Carlo) posterior probabilities were also estimated. 

## 3. Results and Discussion

### 3.1. Genetic Variation in Forage Pea (P. sativum *ssp.* arvense *L.*) Landrace Accessions Using SSR Primers

The genetic variation for 28 SSR loci was calculated based on the number of alleles found (Na), major allele frequency (Maf), observed heterozygosity (Obs-Het), expected heterozygosity (Exp-Het), gene diversity (GD), Nei’s expected heterozygosity (h), effective number of alleles (ne), Shannon’s Information index (I), and polymorphic information content (PIC) among the 61-forage pea (*P. sativum* ssp. *arvense* L.) landrace accessions (Table 3). 

In this study, all primers yielded good or excellent polymorphic band profiles. The 28 SSR loci were confirmed to have a total of 82 alleles in the 61-forage pea (*P. sativum* ssp. *arvense* L.) landrace accessions. The number of alleles per polymorphic locus varied between 2 (PB14, PSAB109, AD100, AA303, AA315, AA-67, AA-321, AD-186, AA-278 and A-9) and 5 (PSAA175 and PSAC58) and the average observed number of alleles per locus was 2.89 (Table 3). When compared to data from comparable studies on other species, these levels of polymorphism indicate that pea is a polymorphic autogamous species [31]. In other words, pea is a self-pollinating species. Smýkal et al. [32] reported that in 25 pea varieties discriminated with SSR markers, the number of alleles ranged from 3 to 6, detecting 38 alleles altogether. Nasiri et al. [33] examined putative duplicate accessions among 20 pea varieties with 57 accessions from wild pisum using 10 out of 20 microsatellite primer pairs to identify genetic relationships in the pisum genus. They discovered 59 alleles in total in the entire dataset, with the PEACPLHPP, AF004843, and AA43090 loci having the most (8 alleles). They also discovered that the wild accessions, PSGAPA 1, PEACPLHPPS, AF004843, and PSmpsaa278c loci all produced 7 alleles, with the number of alleles per locus ranging from 2 to 8, with a mean of 5.9 alleles per locus. Demirkol and Yilmaz [15] showed that 51 genotypes were successfully discriminated using 32 SSR markers, with 127 alleles detected and the number of alleles per primer ranging from 2 to 7 with an average of 3.97, which is higher than the value obtained in our study. All primers were determined to be polymorphic. Zhuang et al. [34] found 37 SSR markers in amplified PCR products, 11 of which produced polymorphism in 23 people, including the parents of recombinant inbred lines, with 2 to 4 alleles in 23 individuals. It is difficult to compare the levels of variety with research because the number of alleles discovered per marker and the genetic diversity of markers are both dependent on the number of genotypes examined [35]. Teshome et al. [14] exposed 37 alleles which were detected across the 12 EST-SSR loci of the 46 field pea accessions genotyped. However, in pea, the mean number of alleles per polymorphic marker was 2.3 [36], 4.5 [37], 3.6 [35], 5 [38], 4 [39], 5 [40], 3.8 [31], 4 [41,42], and 3.8 [32].

In our study, the average MAF was 0.60, with a range of 0.37 (PSAC58 and PSAD280) to 0.98 (AA-278 and A-9) (Table 3). Additionally, the average observed heterozygosity (Obs-He) among all the SSR markers used was 0.22, ranging from zero (AA303, AA315, AA-67, AA-321, AD-186, AA-278 and A-9) to 0.85 (PB14) (Table 3). 

The average expected heterozygosity (Exp-He) was 0.50 with a minimum value of 0.03 (A-9) and a maximum of 0.76 (PSAC58) (Table 3), which indicated an excess of homozygosity within populations, which could be due to inbreeding. Differences in Obs-He could be due to a variety of factors, including the molecular markers utilized. The average Obs-He varied depending on a number of factors, including the number of selections and wild origins, as well as the sample locations’ geographic location (Erzurum, Kars, Bayburt, Bingol, and Gumushane). The broad geographical regions of the aggregation areas, the great number of local species analyzed, and the large number of landraces studied may have contributed to the high number of alleles per locus and identified in our study. Except for AB53 and PVSBE2, observed heterozygosity was lower than expected, according to Sarikamis et al. [43]. This is most likely owing to the pea’s inbreeding nature, rather than null alleles. Zhuang et al. revealed expected heterozygosity values ranging from 0 to 0.43 and 0.31 to 0.83, respectively [34].

The average GD was 0.49, with a range of 0.03 (AA-278 and A-9) to 0.74 (PSAC58). The means of number of effective alleles, which was 2.15, varied between 1.03 (A-9) and 4.07 (PSAC58) (Table 3). Nei’s expected heterozygosity with a mean of 0.49 was at the lowest (0.03) A-9 locus; the highest (0.75) was observed in the PSAC58 locus (Table 3). In this study, Shannon’s Information index, which ranged from 0.08 (A-9) to 1.47 (PSAC58), was found to be an average of 0.81 (Table 3). Teshome et al. [14] showed that the observed number of alleles (na) per locus ranged from two to six. The effective number of alleles (ne) at each locus ranged from 1.05 to 2.83. Observed heterozygosity (Ho) varied from zero to 0.05. The Shannon diversity index (I) per locus ranged from 0.11 to 1.13. The average Shannon diversity index for all loci was 0.53.

The PIC value indicates the discriminating power of the marker. The average PIC value was 0.41, with a range of 0.03 (AA-278 and A-9) to 0.70 (PSAC58). The PIC value provides a clear picture for diversity assessment as it takes into account the relative frequencies of each available band [44]. Smýkal et al. [18] reported that 25 pea varieties discriminated with SSR markers had PIC ranging from 0.10 to 0.75, on average 0.52, which is higher than the value obtained in our study. Nasiri et al. [33] showed that the polymorphism information content (PIC) in their entire collection varied from 0.556 to 0.839 and averaged 0.72. Demirkol and Yilmaz [15] showed that the average polymorphism information content (PIC) value was 0.63, ranging from 0.17 to 0.89. Loridon et al. [31] reported mean PIC of 0.62, Haghnazari et al. [40] 0.53, Smýkal et al. [32] 0.52, Smýkal et al. [18] 0.89, Cupic et al. [37] 0.51, and Gong et al. [36] 0.41. Furthermore, the average PIC and allele values found in this study were greater than those found in Handerson et al. [45] and Jain et al. [46] and were comparable to those found in refs. [13,37,47,48,49,50,51,52]. In this study, however, average PIC and allele values were found to be lower than those found by [18] and [53]. The PIC and MAF value indicate considerable genetic variation among all forage pea (*P. sativum* ssp. *arvense* L.) landraces used. Using a number of molecular markers, genetic variation between individuals in a population or between populations in a crop species, which is produced from genes, environmental impacts, or both, may be easily assessed. In general, the genetic diversity detectable by molecular markers in plants depends on the mode of reproduction, the history of domestication, and the size of the samples analyzed.

### 3.2. Genetic Distance, Cluster Analysis, Molecular Variance Analysis (AMOVA), and Principal Coordinate Analysis (PCoA) for SSR Markers

We analyzed phylogenetic relationships for the 61-forage pea (*P. sativum* ssp. *arvense* L.) accessions used in our study using the 28 SSR markers in order to better understand genetic variability and genetic relationships among accessions of the forage pea (*P. sativum* ssp. *arvense* L.) landraces used in our study. The forage pea (*P. sativum* ssp. *arvense* L.) landrace accessions aggregated into three primary groups with a genetic dissimilarity variation of 0.10 (G30 and G34) and 0.66 (G1 and G32), according to a phylogenetic tree generated using UPGMA and 28 SSR markers (Figure 2). Differences between accessions at the time of recognition could account for the increased calculated distance. Group I contained one accession of the forage pea (*P. sativum* ssp. *arvense* L.) landraces including G13. Group II contained 59 accessions of the forage pea (*P. sativum* ssp. *arvense* L.) landraces. In addition, Group II was divided into three subclusters. The first subcluster (G II-1) contained 31 accessions including G29, G44, G43, G42, G41, G35, G39, G38, G40, G37, G31, G30, G32, G19, G21, G20, G14, G18, G17, G16, G15, G24, G34, G33, G28, G26, G25, G27, G22, G12, and G11. The second subcluster (G II-2) contained two accessions including G45 and G8. The third subcluster (G II-3) contained 26 accessions including G47, G60, G61, G59, G10, G49, G58, G57, G54, G53, G52, G51, G50, G48, G56, G55, G46, G23, G7, G36, G6, G5, G4, G3, G9, and G2 (Figure 2). Briefly, out of G1 (cluster III) and G13 (Cluster I), the rest of the landraces were Cluster III. Thus, most of the accessions collected from various provinces were clustered in the same group; mixing and grouping was not clear according to geographic region. The non-relatedness of landraces from the same region could be due to climatic conditions. Cupic et al. [37] found that the average distance between two groups of accessions was 0.69 when they looked at the genetic diversity of 18 pea (*Pisum sativum* L.) genotypes using pedigree, morphological, and 26 SSR molecular data. In addition, they discovered that three main clusters (at value 0.61 of distance) as well as two sub-clusters within the sativum group (at value 0.49 of distance) were composed of dendrogram-confirmed groups of *arvense* and *sativum* accessions. Nasiri et al. [33], investigating the genetic diversity among 77 accessions of pea (*Pisum sativum* L.) based on 10 SSR markers, showed that cluster analysis and principal coordinate analysis located accessions in three groups: their result was similar to ours.

The UPGMA cluster analysis and PCOA using genetic markers revealed that all genotypes were not clearly differentiated from the 61-forage pea (*P. sativum* ssp. *arvense* L.) landrace accessions. According to our results, the clustering patterns could not clearly distinguish the forage pea (*P. sativum* ssp. *arvense* L.) landrace accessions by province of collection. This clustering of genotypes showed that there was no significant relationship between geographic origin and genetic similarity. This result indicates that there is a certain level of gene flow between genotypes. Cupic et al. [37] found that the genetic distance across pea accessions might range from 0.24 to 0.84 based on SSR markers. Differences between accessions at the time of recognition may have resulted in a larger calculated distance.

Molecular analysis of variance (AMOVA) results on forage pea (*P. sativum* ssp. *arvense* L.) landrace accessions to assess variation in populations showed that the within-populations (82.00%) were higher than the between-populations (18.00%) (Table 4). Analysis of molecular variance (AMOVA) demonstrated high variability within arvense groups, indicating that there is still sufficient variety for them to be used in breeding schemes. The variance among the population is low. However, the overall comparison showed significant variation between accessions due to differences in the frequency of multiple and unique and rare alleles between accessions. Because Ardahan Province is regarded as one of the primary forage peas growing locations, the grouping observed in our study can be justified by this assumption. Diffusion of seeds may also occur among provinces through farmers. Another possible reason is that a large portion of the studied material was collected from Ardahan Province, which led to the clustering of these landraces close to each other. This result indicates that there is gene flow between populations [54]. MW Blair, A Soler, and AJ Cortes [55], in their study with 36 SSR markers in 104 wild bean genotypes, determined that the variance within the populations was 98%. Dissimilarity to our findings was observed in Nasiri et al. [33] in wild species accessions of pea (*Pisum sativum* L.) using SSR markers studies—the intergroups component of variance (29%) being lower than the intragroup component of variance (71%). Furthermore, Teshome et al. [14] reported that the genetic differentiation among accessions was 41% while the variation within was 59%. The explanation for this could be that the region contains a variety of altitudes and climates, which tend to increase diversity. Analysis of molecular variance (AMOVA) demonstrated high variability among population groupings, indicating that there is still sufficient variety to be employed in breeding efforts.

Principal coordinate analysis (PCoA) is a multivariate dataset that provides the ability to find and archive key patterns in multiple loci and multiple samples. With this technique, the distances between the groups, which are formed thanks to the two-dimensional diagram formed by the similarity or distance matrix between the individuals, reflect the real distances [56]. PCoA is used to provide a spatial representation of the relative genetic distances between populations [57]. SSR-based clustering was found to be beneficial in distinguishing forage pea (*P. sativum* ssp. *arvense* L.) landraces depending on their origins. There was no correlation between PCoA grouping and cluster analysis in this study. (Figure 3). In our study, the PCoA was performed using the neutral genetic distance of Nei. The percentage of genetic diversity explained by each of the three main coordinates of the basic coordinate analysis was determined to be 21.59, 16.67, and 13.28, respectively, and these first three components explained 51.54% of the diversity (Table 5). Although the groups were not completely separated in the 2D diagram obtained over the first two components, the distribution of genotypes on the diagram indicated the presence of genetic diversity (Figure 3).

### 3.3. Population Structure among 61 Forage Peas (P. sativum *ssp.* arvense *L.*) Landrace Accessions Using SSR

To better understand the population structure of the 61-forage pea (*P. sativum* ssp. *arvense* L.) landraces, we used the model-based approach in STRUCTURE software to divide each accession into matching subgroups. For the 61 landraces of forage pea (*P. sativum* ssp. *arvense* L.), the highest value of DK was K = 3. (Figure 4). As a consequence, at K = 3 (membership probability 0.8), all accessions were sorted into three main groups and an admixed group (red [A], green [B], blue [C]) (Figure 5).

At K = 3, Group I included 14 accessions of the forage pea (*P. sativum* ssp. *arvense* L.) landraces including G26, G31, G28, G25, G34, G40, G39, G30, G33, G37, G35, G22, G27, and G20. The genotypes of this group include some of the Ardahan genotypes, and all of the Kars and Giresun genotypes. Group II included 18 accessions of the forage pea (*P. sativum* ssp. *arvense* L.) landraces counting G57, G56, G52, G51, G55, G50, G54, G61, G46, G48, G59, G53, G58, G47, G49, G44, G45, and G60. The genotypes of this group included only the rest of the Ardahan province genotypes. Group III included nine accessions of the forage pea (*P. sativum* ssp. *arvense* L.) landraces: G7, G6, G4, G5, G9, G10, G3, G2, and G8. Additionally, the genotypes of this group comprise the Erzurum province genotypes except G1, and half of the Bayburt province genotypes. The admixed group contained a total of 20 accessions of the forage pea (*P. sativum* ssp. *arvense* L.) landraces comprising G12, G32, G21, G38, G41, G19, G14, G29, G24, G43, G60, G42, G18, G23, G36, G13, G15, G16, G17, G11, and G1. The genotypes of this group cover all the provinces examined in the study. Our findings revealed that, despite being gathered from various geographical and topographic regions throughout Turkey, there was no evidence of a place of origin association between these inclusions. According to Baloch et al. [13], population structure analysis divided the genotypes into three sub-populations at K = 3 and additional division occurred at K = 5, indicating that all landraces are genetically descended from five subpopulations. Based on the 37 alleles dispersed among 12 EST-SSR loci, Teshome et al. [14] deduced that (K = 9) populations can include all individuals from the 46 accessions with the highest likelihood. Wang et al. [58] identified three subpopulations among 256 pea genotypes. Although it was primarily modeled to measure the quantity of allelic fixation due to genetic alteration, the forage pea (*P. sativum* ssp. *arvense* L.) landrace sub-populations varied for F_st_ (a measure of population differentiation due to genetic structure), which was determined to observe the relation within alleles drawn at varying scales of a hierarchically sub-divided population. Non-differentiation and perfect differentiation between an initial population and its sub-populations are shown by F_st_ values of 0 and 1, respectively. The F_st_ ranges from 0 to 0.05 show little genetic differentiation, while the ranges from 0.05 to 0.15, 0.15 to 0.25, and above 0.25, respectively, indicate moderate, large, and very substantial genetic differentiation [59]. According to these data, the F_ST_ (F-statistic) value was determined to be 0.2351, 0.3838, and 0.2506 in the first, second, and third sub-populations, respectively, and the mean F_st_ (F-statistic) value of 0.2898 confirmed the segregation of all subpopulations and the diversity in SSR alleles (Table 6). For genetic differentiation based on F_st_ values among three forage pea sub-populations, sub-populations B and C were found to be the most diverse populations with a value of 0.1087 (Table 6 and Table 7). However, genetic variation in subpopulations A, B, and C in our study was moderate, very large, and large respectively. Expected heterozygosity values varied between 0.2581 (sub-population B) and 0.3223 (sub-population C) with an average value of 0.2924 (Table 6).

## 4. Conclusions

In northeastern Anatolia, especially in Erzurum, Bayburt, Ardahan, Kars, and Giresun, fodder pea is a plant that has been grown for many years both for hay and for animal feed grains. The straw, grain, and straw of fodder peas are a source of food and energy for livestock and are an alternative to barley and vetch in animal nutrition. Due to the superior properties of fodder peas, many studies in Turkey have made great efforts in recent years to develop new high-yielding varieties using local or introduced fodder pea materials and emphasized the diversity of pea genotypes based on agro-morphological traits, micro and macro nutrients and morphological markers, and protein mineral content [60,61,62,63,64]. Although fodder pea genotypes collected from the eastern Black Sea region have previously been characterized, fodder pea ecotypes from the northeastern Anatolia region of Turkey have not been genetically characterized before. In addition, there have been few studies on forage pea germplasm native species in Turkey. Therefore, these are important first steps to a better understanding and maintenance of the forage pea germplasm of the region. Genomic SSRs, selected on the basis of high polymorphism information content, successfully assisted in the differentiation of genotypes in this study, resulting in successful amplifications of expected sizes. To the best of our knowledge, few studies have investigated genetic diversity and association among Turkish forage pea native species using SSR markers.

Consequently, this study focusing on local fodder pea varieties collected from the northeastern Anatolia region of Turkey was evaluated at the molecular level with the SSR marker system. Landraces act as sources of new genes and represent higher intra- and inter-variation that increase their importance in breeding programs. Recent efforts to sample plant genetic resources from farmers will result in greater diversity for use in breeding. The collection of crop biodiversity is often associated with native species grown in non-industrial agricultural areas. This study reveals the importance of taking inventory of local native varieties. In addition, through marker-assisted selective breeding programs, they can provide support for the effective selection and use of available accessions, allowing better participation selection in molecular breeding programs for the development of fodder pea. In addition, for the long-term breeding program of forage pea, there is a need to convert the identified genes into the Kompetitive Allele Specific PCR assay. The results of this study indicate genetic inbreeding among Turkish fodder pea germplasm grown in the northeastern Anatolia region of Turkey.

## Figures and Tables

**Figure 1 genes-13-01086-f001:**
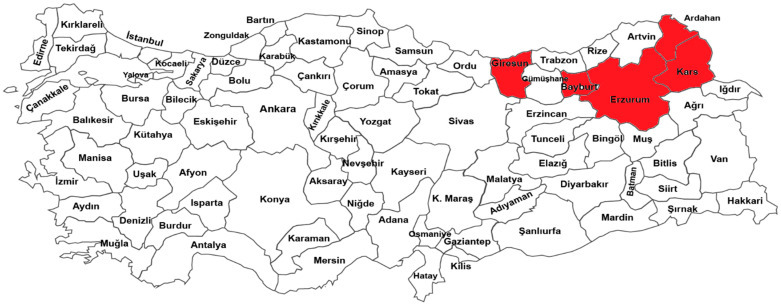
Geographic distribution of forage pea landraces collected from various geographical provinces in the northeastern Anatolia region of Turkey.

**Figure 2 genes-13-01086-f002:**
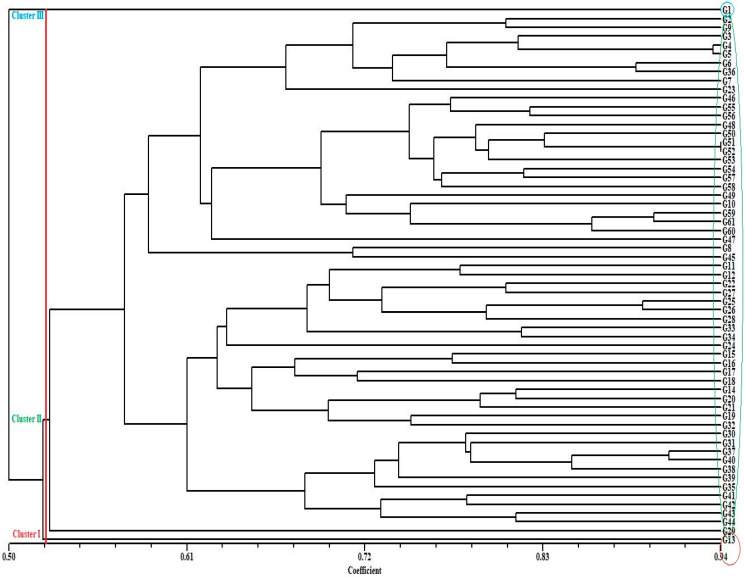
UPGMA dendrogram of 61 forage peas (*P. sativum* ssp. *arvense* L.) landrace accessions based on 28 SSR markers.

**Figure 3 genes-13-01086-f003:**
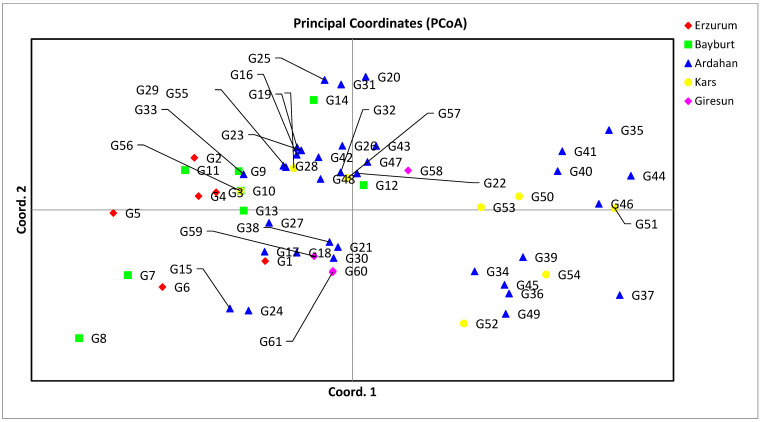
PCoA of forage pea (*P. sativum* ssp. *arvense* L.) landrace accessions.

**Figure 4 genes-13-01086-f004:**
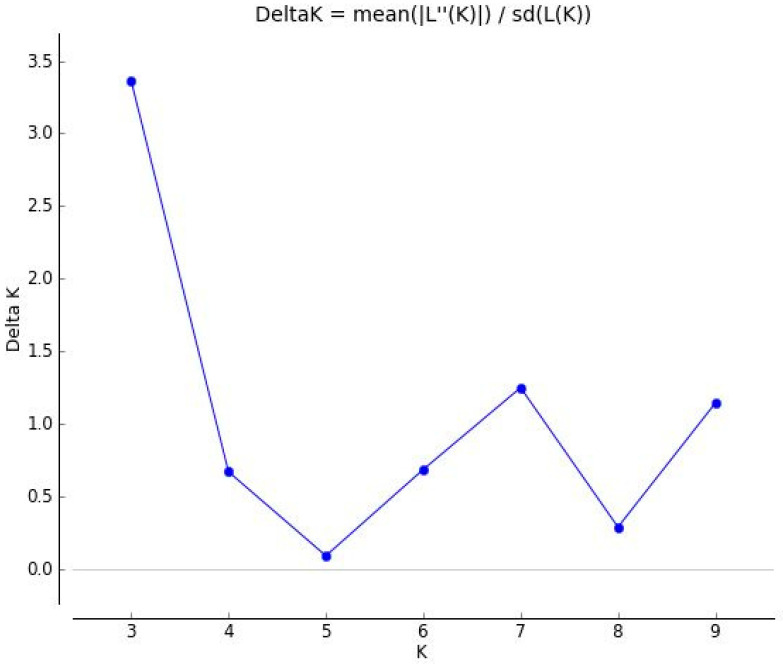
Delta K value proposing the presence of three populations of forage pea genotypes using iPBS-retrotransposons marker system.

**Figure 5 genes-13-01086-f005:**
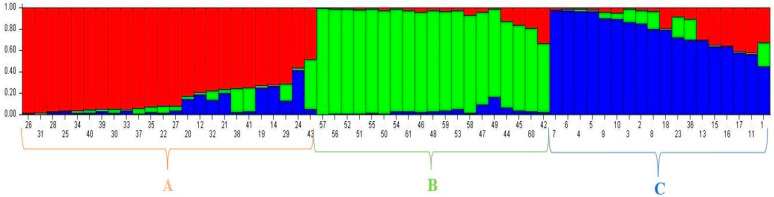
The population structure pattern for the highest ΔK value, K = 3, of 61 forage pea (*P. sativum* ssp. *arvense* L.) landrace accessions based on 28 SSR markers.

**Table 1 genes-13-01086-t001:** List of forage pea (*P. sativum* ssp. *arvense* L.) by collection and coordinates.

Code Number	Association No	Location	Latitude	Longitude	Altitude (m)
G1	Ovaçevirme-1	Erzurum	39.19.57 N	41.47.51 E	1586
G2	Ovaçevirme-2	Erzurum	39.19.57 N	41.47.51 E	1586
G3	Ovaçevirme-3	Erzurum	39.19.57 N	41.47.51 E	1586
G4	Ovaçevirme-4	Erzurum	39.19.57 N	41.47.51 E	1586
G5	Ovaçevirme-5	Erzurum	39.19.57 N	41.47.51 E	1586
G6	Şenkaya Merkez	Erzurum	40.33.27 N	42.20.37 E	1826
G7	İncili-1	Bayburt	40.23.24 N	40.12.16 E	1579
G8	İncili-2	Bayburt	40.23.22 N	40.12.14 E	1578
G9	İncili-3	Bayburt	40.23.22 N	40.12.14 E	1575
G10	Çiğdemtepe	Bayburt	40.19.50 N	40.08.16 E	1564
G11	Arpalı	Bayburt	40.21.56 N	40.06.17 E	1538
G12	Aşağıkırzı	Bayburt	40.23.23 N	40.08.48 E	1535
G13	Değirmencik-1	Bayburt	40.21.29 N	40.14.30 E	1532
G14	Değirmencik-2	Bayburt	40.21.29 N	40.14.30 E	1532
G15	Ardahan Merkez-1	Ardahan	41.09.45 N	42.56.15 E	1905
G16	Ardahan Merkez-2	Ardahan	41.09.45 N	42.56.15 E	1905
G17	Ardahan Merkez-3	Ardahan	41.09.45 N	42.56.15 E	1905
G18	Çamlıçatak-1	Ardahan	41.07.42 N	42.49.59 E	1798
G19	Çamlıçatak-2	Ardahan	41.07.42 N	42.49.59 E	1798
G20	Sulakyurt	Ardahan	41.09.50 N	42.36.59 E	1920
G21	Döşeli-1	Ardahan	41.08.59 N	42.44.45 E	1815
G22	Döşeli-2	Ardahan	41.08.59 N	42.44.45 E	1815
G23	Kartalpınar	Ardahan	41.08.59 N	42.44.45 E	2063
G24	Tepeler	Ardahan	41.03.51 N	42.34.30 E	2044
G25	Oburcak	Ardahan	41.20.59 N	42.52.10 E	2100
G26	Seyitören	Ardahan	41.23.18 N	42.48.35 E	2224
G27	Serhat	Ardahan	41.20.31 N	42.85.11 E	1950
G28	Cumhuriyet	Ardahan	41.20.57 N	42.52.16 E	1922
G29	Burmadere	Ardahan	41.18.09 N	42.47.05 E	2044
G30	Tepeköy	Ardahan	41.20.50 N	42.48.26 E	2076
G31	Selamverdi	Ardahan	41.14.06 N	42.51.44 E	1810
G32	Ortakent	Ardahan	41.13.45 N	42.56.05 E	1819
G33	Yamçılı	Ardahan	42.20.29 N	42.80.12 E	2034
G34	Koyunpınarı	Ardahan	41.14.20 N	42.46.40 E	1860
G35	Çayağzı	Ardahan	41.11.34 N	42.51.02 E	1813
G36	Avcılar	Ardahan	41.15.02 N	42.49.33 E	1801
G37	Öncül	Ardahan	41.14.18 N	43.09.59 E	1793
G38	Aşağıcambaz	Ardahan	41.14.41 N	43.24.10 E	2044
G39	Eşmepınar	Ardahan	41.06.50 N	43.09.07 E	1889
G40	Eskibeyrehatun	Ardahan	41.07.42 N	42.57.30 E	2040
G41	Kaşlıkaya	Ardahan	41.09.38 N	43.04.12 E	1857
G42	Sazlısu	Ardahan	41.06.16 N	43.06.58 E	1913
G43	Kenarbel	Ardahan	41.11.26 N	43.09.53 E	1788
G44	Tahtakıran	Ardahan	40.30.32 N	42.35.23 E	2042
G45	Dedekılıç	Ardahan	40.51.45 N	42.33.31 E	2040
G46	Yiğitkonağı	Ardahan	40.57.16 N	42.34.04 E	2048
G47	Çayırbaşı-1	Ardahan	40.52.24 N	43.37.56 E	2045
G48	Çayırbaşı-2	Ardahan	40.52.24 N	43.37.56 E	2045
G49	Balçeşme	Ardahan	40.49.54 N	42.49.29 E	2165
G50	Subatan	Kars	40.36.12 N	43.26.07 E	1788
G51	Doğruyol	Kars	41.03.00 N	43.20.10 E	2031
G52	Paslı	Kars	40.17.05 N	42.57.32 E	1838
G53	Yolgeçmez	Kars	40.25.07 N	42.45.11 E	1898
G54	Iğdır	Kars	40.21.12 N	42.45.11 E	1815
G55	Gölbaşı	Kars	40.76.57 N	42.98.47 E	2015
G56	Yukarışallıpınar-1	Kars	40.43.63 N	42.61.42 E	2081
G57	Yukarışallıpınar-2	Kars	40.43.63 N	42.61.42 E	2081
G58	Sarvan	Giresun	40.53.03 N	38.27.19 E	266
G59	Görele-1	Giresun	41.01.54 N	39.00.04 E	24
G60	Görele-2	Giresun	41.01.54 N	39.00.04 E	24
G61	Giresun Merkez	Giresun	40.52.07 N	38.23.38 E	18

**Table 2 genes-13-01086-t002:** Twenty-eight SSR primers for genetic diversity analysis among forage pea (*P. sativum* ssp. *arvense* L.) landrace accessions.

Primer No.	Marker Name	Forward (5′–3′)	Backward (5′–3′)
1	PB14	GAGTGAGCTTTTTAGCTTGCAGCCT	TGCTTGAGAACAGTGACTCGCA
2	PSAA18	CTGTAGACCAAGCCCAAAAGAT	TGAGACACTTTTGACAAGGAGG
3	PSAA175	TTGAAGGAACACAATCAGCGAC	TGCGCACCAAACTACCATAATC
4	PSAC58	TCCGCAATTTGGTAACACTG	CGTCCATTTCTTTTATGCTGAG
5	PSAC75	CGCTCACCAAATGTAGATGATAA	TCATGCATCAATGAAAGTGATAAA
6	PSAA219	ATTTGTGCAATTGCAATTTCATT	CGAAAACGCTTTGCATCCTA
7	PSAD83	CACATGAGCGTGTGTATGGTAA	GGGATAAGAAGAGGGAGCAAAT
8	PSAD270	CTCATCTGATGCGTTGGATTAG	AGGTTGGATTTGTTGTTTGTTG
9	PSAA456	TGTAGAAGCATAAGAGCGGGTG	TGCAACGCTCTTGTTGATGATT
10	PSAB23	TCAGCCTTTATCCTCCGAACTA	GAACCCTTGTGCAGAAGCATTA
11	PSAB47	TCCACAATACCATCTAAATGCCA	AATTTGTTCAGTTGAAATTTCGTTTC
12	PSAA497	TTGTGACTGATTTAGAAGTTTCCCAC	TTGATGAGTTGCAATTTCGTTTC
13	PSAD280	TGGTGCTCGTGATTAATTTCACATA	ACTAAACAACCAACTGCCAAAACTG
14	PSAB72	ATCTCATGTTCAACTTGCAACCTTTA	TTCAAAACACGCAAGTTTTCTGA
15	PSAB109	GAACCCTTGTGTAGAAGCATTTGTG	GAGCTACTGTGAGTCTGATGCCATTAT
16	PSAB141	ATCCCAATACTCCCACCAATGTT	AGACTTAGGCTTCCCTTCTACGACTT
17	PSAB161	CTCAAGTGAAGACTTGGAATTTCGTT	TTTGGTCTTCCTCAAGTGATAAGATG
18	AD100	TACACCCAAGACGACAAGCCT	GGAGCTTCCGCTTGATTCTCT
19	AD134	TTTATTTTTCCATATATTACAGACCCG	ACACCTTTATCTCCCGAAGACTTAG
20	AA303	GGGTGAAGGAAAATCGTGA	GCATCCCATAAAATTGGTTCT
21	AA315	AGTGGGAAGTAAAAGGTGTAG	TTTCACTAGATGATATTTCGTT
22	AA-67	CCCATGTGAAATTCTCTTGAAGA	GCATTTCACTTGATGAAATTTCG
23	AA-321	CTGCAGCCTGTACAAGTGG	CCAGTTACAAGATTGATGTTTATGG
24	AD-186	TCAATGACGTGTTGATCGAGGA	CCATGCTTTGCACCGAAAGTAA
25	AA-278	CCAAGAAAGGCTTATCAACAGG	TGCTTGTGTCAAGTGATCAGTG
26	A-9	GTGCAGAAGCATTTGTTCAGAT	CCCACATATATTTGGTTGGTCA
27	AD-141	AATTTGAAAGAGGCGGATGTG	ACTTCTCTCCAACATCCAACGA
28	AD-237	AGATCATTTGGTGTCATCAGTG	TGTTTAATACAACGTGCTCCTC

**Table 3 genes-13-01086-t003:** Genetic diversity indices of forage pea (*P. sativum* ssp. *arvense* L.) landrace accessions.

Marker	Na	Maf	Ob-Het	Ex-Het	GD	h	ne	I	PIC
PB14	2.00	0.57	0.85	0.49	0.49	0.49	1.96	0.68	0.37
PSAA18	4.00	0.43	0.41	0.62	0.62	0.62	2.60	1.07	0.54
PSAA175	5.00	0.57	0.42	0.62	0.62	0.62	2.62	1.20	0.58
PSAC58	5.00	0.37	0.49	0.76	0.74	0.75	4.07	1.47	0.70
PSAC75	3.00	0.51	0.58	0.61	0.60	0.60	2.52	0.99	0.52
PSAA219	4.00	0.60	0.04	0.56	0.54	0.54	2.19	0.97	0.48
PSAD83	3.00	0.44	0.13	0.63	0.62	0.62	2.66	1.03	0.55
PSAD270	3.00	0.53	0.33	0.56	0.60	0.55	2.24	0.91	0.52
PSAA456	3.00	0.59	0.13	0.52	0.51	0.51	2.05	0.81	0.42
PSAB23	3.00	0.45	0.53	0.65	0.64	0.64	2.78	1.06	0.57
PSAB47	3.00	0.53	0.44	0.60	0.60	0.60	2.48	0.99	0.52
PSAA497	3.00	0.80	0.23	0.34	0.33	0.34	1.51	0.55	0.28
PSAD280	3.00	0.37	0.08	0.67	0.66	0.66	2.97	1.09	0.59
PSAB72	3.00	0.48	0.22	0.62	0.61	0.62	2.60	1.01	0.53
PSAB109	2.00	0.75	0.00	0.38	0.38	0.38	1.60	0.56	0.30
PSAB141	3.00	0.52	0.09	0.54	0.54	0.54	2.16	0.84	0.43
PSAB161	3.00	0.52	0.19	0.55	0.55	0.55	2.20	0.86	0.45
AD100	2.00	0.54	0.00	0.51	0.50	0.50	2.00	0.69	0.37
AD134	3.00	0.65	0.04	0.50	0.47	0.50	1.99	0.79	0.38
AA303	2.00	0.67	0.00	0.44	0.44	0.44	1.79	0.63	0.34
AA315	2.00	0.88	0.00	0.22	0.22	0.22	1.27	0.37	0.19
AA-67	2.00	0.56	0.00	0.50	0.49	0.49	1.97	0.69	0.37
AA-321	2.00	0.89	0.00	0.22	0.20	0.21	1.27	0.37	0.18
AD-186	2.00	0.53	0.00	0.50	0.50	0.50	1.99	0.69	0.37
AA-278	2.00	0.98	0.00	0.15	0.03	0.15	1.18	0.29	0.03
A-9	2.00	0.98	0.00	0.03	0.03	0.03	1.03	0.08	0.03
AD-141	3.00	0.67	0.06	0.47	0.46	0.46	1.86	0.73	0.38
AD-237	4.00	0.57	0.77	0.61	0.60	0.60	2.51	1.11	0.55
Mean	2.89	0.60	0.22	0.50	0.49	0.49	2.15	0.81	0.41

Na: Observed Number of alleles, Maf: major allele frequency, Obs-Het: observed heterozygosity, Exp-Het: Expected heterozygosity, GD: Gene diversity, h: Nei’s expected heterozygosity, ne: Effective number of alleles, I: Shannon’s Information index, PIC: Polymorphism information content.

**Table 4 genes-13-01086-t004:** AMOVA of forage pea (*P. sativum* ssp. *arvense* L.) landrace accessions.

Source	Degree of Freedom (DF)	Sum of Squares (SS)	Variance Component	% Of Total Variance	*p*-Value
Among Population	4	78.52	1.40	18	0.18
Within Population	56	351.06	6.26	82	0.001
Total	60	429.59	7.67	100	-

**Table 5 genes-13-01086-t005:** PCoA analysis of forage pea (*P. sativum* ssp. *arvense* L.) landrace accessions.

Axis	1	2	3
%	21.59	16.67	13.28
Cum %	21.59	38.26	51.54

**Table 6 genes-13-01086-t006:** Heterozygosity and Fst values of forage pea (*P. sativum* ssp. *arvense* L.) landrace accession sub-populations.

Sub-Population (K)	Expected Heterozygosity	Fst Value
A	0.2969	0.2351
B	0.2581	0.3838
C	0.3223	0.2506
Mean	0.2924	0.2898

**Table 7 genes-13-01086-t007:** Genetic differentiation based on Fst values among forage pea (*P. sativum* ssp. *arvense* L.) landrace accession sub-populations identified by population structure analysis.

Sub-Populations (K)	Sub-Population A	Sub-Population B	Sub-Population C
Sub-population A	-	0.1024	0.0690
Sub-population B	0.1024	-	0.1087
Sub-population C	0.0690	0.1087	-

## Data Availability

Data is contained within the article.

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
