# Peer review of "SSR-Based Molecular Identification and Population Structure Analysis for Forage Pea (Pisum sativum var. arvense L.) Landraces"

_genes, 2022, doi:10.3390/genes13061086_

Round 1

Reviewer 1 Report

Dear authors

After a revision of the article entitled "SSR-Based Molecular Identification and Population Structure Analysis for Forage Pea (Pisum sativum var. arvense L.) Landraces" under the number genes-1755733. I gave an unfavorable opinion but if it will be improved could be accepted after Major Revision.

- I did not reject the article directly because I find that the authors used a large number of SSR primers and above all they worked with polyacrylamide gels, I know the details of the work very well.

but at the same time I gave an unfavorable opinion because there is no originality and the whole article is written and all the methodologies are old there are no novelties.

Also I think the article could be appropriate in a National newspaper because the samples from some regions in Turkey so I suggest to expand the provenances from several regions of Turkey.

- you can put an example of amplification on acrylamide gel to enhance your work and show the qualities of your gels.

- for the PIC values ​​there are primers which are not effective in discriminating between your individuals, I think which is useless to use them later.

An English revision is necessary.

Good Luck

Author Response

Dear Editor,

Thank you for your valuable comments about our manuscript "SSR-Based Molecular Identification and Population Structure Analysis for Forage Pea (Pisum sativum var. arvense L.) Landraces to Genes-1755733- Open Access Journals. We revised our article according to the comments of the editors and the referees. Changes made in the text were highlighted yellow. Responses to comments were listed below.

Thank you for reconsidering our manuscript for publication.

Comments

Response

1) You can put an example of amplification on acrylamide gel to enhance your work and show the qualities of your gels.

Figure was added to Supplementary section.

2) For the PIC values ​​there are primers which are not effective in discriminating between your individuals, I think which is useless to use them later.

PIC values for co-dominant markers range from 0 to 1. The mean of PIC values in our study was 0.41 with a range of 0.03–0.70. In our study 11 out of 28 number of SSR markers have PIC value higher 0.50 value.

Reviewer 2 Report

This study evaluated the diversity of forage pea specimens, collected from the northeastern Anatolia region of Turkey using simple sequence repeat (SSR) markers. The results suggested that SSR markers could be constantly used to illuminate the genetic diversity of forage pea landraces and can potentially be incorporated into future studies that examine the diversity within a larger collection of forage pea genotypes from diverse regions. However, the followings need to be clarified.

 Major revision

1. The abstract needs to be summarized and concise to be more readable.

2. G-SSR is derived from genomic sequences and rarely locates genes. EST-SSR is derived from transcriptome sequences and can be mapped to related functional genes. Why not adopt EST-SSR, it needs to be clearly explained.

 Minor revision

1. Consistency in article format.

2. Keep pictures beautiful, legible, clarity and normative.

2. Maintain a standardized and uniform reference format.

Author Response

Dear Editor,

Thank you for your valuable comments about our manuscript "SSR-Based Molecular Identification and Population Structure Analysis for Forage Pea (Pisum sativum var. arvense L.) Landraces to Genes-1755733- Open Access Journals. We revised our article according to the comments of the editors and the referees. Changes made in the text were highlighted yellow. Responses to comments were listed below.

Thank you for reconsidering our manuscript for publication.

Comments

Response

1) The abstract needs to be summarized and concise to be more readable.

We edited the abstract according your suggest.

2) G-SSR is derived from genomic sequences and rarely locates genes. EST-SSR is derived from transcriptome sequences and can be mapped to related functional genes. Why not adopt EST-SSR, it needs to be clearly explained?

1.       EST-SSRs belong to the transcribed regions of DNA. Although, the value of EST-SSRs when compared to genomic SSRs is enhanced by several factors including high transferability, potential to attribute function to genes affecting traits of interest, EST-derived SSR markers are generally less polymorphic than genomic SSRs [Chen H, Qiao L, Wang L, Wang S, Blair MW, Cheng X. Assessment of genetic diversity and population structure of mung bean (Vigna radiata) germplasm using EST-based and genomic SSR markers. Gene. 2015; 566(2): 175–183].

2.       Though significant efforts have been made to develop SSR markers in recent years for many crops, no more than 100 polymorphic SSR markers have been detected and published for forage pea, with most of them being gSSRs due in part to low diversity levels. Polymorphic eSSR markers are presently rare for forage pea, which hinders its genetic characterization.

3.       However, very few molecular markers have been found which were linked to a desirable gene locus in legume crops to date, especially for forage pea.

3) Consistency in article format.

The manuscript has been revised according to the format of the journal.

4) Keep pictures beautiful, legible, clarity and normative.

The all pictures have been revised according to the format of the journal.

5) Maintain a standardized and uniform reference format.

The references have been revised according to the format of the journal.

Reviewer 3 Report

I have some minor correction suggestions in the text. These are so insignificant that they do not disturb the semantic integrity of the presentation. You can see these recommendations in the attached pdf file.

There are different methods for the characterization of genetic resources. Although it is expected to use all methods (molecular, morphological, biochemical, etc.) together in a single study, this is not always possible.

In such cases, it would be helpful to review other characterization studies conducted in the same geography and compare the results with these data.

Other recent studies on the characterization of Turkey's local Pisum sativum L. resources can be added to the "discussion" section.

For example:

1- Protein and mineral contents of pea (Pisum sativum L.) genotypes grown in Central Anatolian region of Turkey

2- Determination of morphological variability of different pisum genotypes using principal component analysis

3-Genetic variability in peas (Pisum sativum L.) from Turkey assessed with molecular and morphological markers

4-Micro and macronutrients diversity in Turkish pea (Pisum sativum) germplasm

5-Characterization of some local pea (Pisum sativum L.) genotypes for agro-morphologicaltraits and mineral concentrations

Author Response

Dear Editor,

Thank you for your valuable comments about our manuscript "SSR-Based Molecular Identification and Population Structure Analysis for Forage Pea (Pisum sativum var. arvense L.) Landraces to Genes-1755733- Open Access Journals. We revised our article according to the comments of the editors and the referees. Changes made in the text were highlighted yellow. Responses to comments were listed below.

Thank you for reconsidering our manuscript for publication.

Comments

Response

Other recent studies on the characterization of Turkey's local Pisum sativum L. resources can be added to the "discussion" section.

For example:

Protein and mineral contents of pea (Pisum sativum L.) genotypes grown in Central Anatolian region of Turkey

Determination of morphological variability of different pisum genotypes using principal component analysis

Genetic variability in peas (Pisum sativum L.) from Turkey assessed with molecular and morphological markers

Micro and macronutrients diversity in Turkish pea (Pisum sativum) germplasm

Characterization of some local pea (Pisum sativum L.) genotypes for agro-morphologicaltraits and mineral concentrations

We added all suggested article in main text and also references were added.
